# Mortality risk prediction of high-sensitivity C-reactive protein in suspected acute coronary syndrome: A cohort study

Amit Kaura[1,2], Adam Hartley[1,2], Vasileios Panoulas[1,2], Ben Glampson[2], Anoop S. V. Shah[1,2,3], Jim Davies[4], Abdulrahim Mulla[2], Kerrie Woods[4], Joe Omigie[5], Anoop D. Shah[6], Mark R. Thursz[2], Paul Elliott[2,7], Harry Hemmingway[6,7], Bryan Williams[6], Folkert W. Asselbergs[6], Michael O'Sullivan[8], Graham M. Lord[9], Adam Trickey[10], Jonathan AC Sterne[11], Dorian O. Haskard[1], Narbeh Melikian[5], Darrel P. Francis[1,2], Wolfgang Koenig[12,13], Ajay M. Shah[5], Rajesh Kharbanda[4], Divaka Perera[9], Riyaz S. Patel[6], Keith M. Channon[4], Jamil Mayet[1,2]*, Ramzi Khamis[1,2]

1 National Heart and Lung Institute, Imperial College London, London, United Kingdom, 2 NIHR Imperial Biomedical Research Centre, Imperial College London and Imperial College Healthcare NHS Trust, London, United Kingdom, 3 London School of Hygiene Tropical Medicine, London, United Kingdom, 4 NIHR Oxford Biomedical Research Centre, University of Oxford and Oxford University Hospitals NHS Foundation Trust, Oxford, United Kingdom, 5 NIHR King's Biomedical Research Centre, King's College London and King's College Hospital NHS Foundation Trust, London, United Kingdom, 6 NIHR University College London Hospitals Biomedical Research Centre, University College London and University College London Hospitals NHS Foundation Trust, London, United Kingdom, 7 Health Data Research, London Substantive Site, United Kingdom, 8 NIHR Cambridge Biomedical Research Centre, University of Cambridge and Cambridge University Hospitals NHS Foundation Trust, Cambridge, United Kingdom, 9 NIHR Manchester Biomedical Research Centre, University of Manchester and Manchester University NHS Foundation Trust, Manchester, United Kingdom, 10 NIHR Bristol Biomedical Research Centre, University of Bristol and University Hospitals Bristol NHS Foundation Trust, London, United Kingdom, 11 NIHR King's Biomedical Research Centre, King's College London and Guy's and St Thomas' NHS Foundation Trust, London, United Kingdom, 12 Deutsches Herzzentrum München, Technische Universität München, Munich, Germany and DZHK (German Centre for Cardiovascular Research), partner site Munich Heart Alliance, Munich, Germany, 13 Institute of Epidemiology and Medical Biometry, University of Ulm, Ulm, Germany

* j.mayet@imperial.ac.uk

## Abstract

### Background

There is limited evidence on the use of high-sensitivity C-reactive protein (hsCRP) as a biomarker for selecting patients for advanced cardiovascular (CV) therapies in the modern era. The prognostic value of mildly elevated hsCRP beyond troponin in a large real-world cohort of unselected patients presenting with suspected acute coronary syndrome (ACS) is unknown. We evaluated whether a mildly elevated hsCRP (up to 15 mg/L) was associated with mortality risk, beyond troponin level, in patients with suspected ACS.

### Methods and findings

We conducted a retrospective cohort study based on the National Institute for Health Research Health Informatics Collaborative data of 257,948 patients with suspected ACS who had a troponin measured at 5 cardiac centres in the United Kingdom between 2010 and

**Data Availability Statement:** We are unable to extract or publish patient level data due to data protection and governance restrictions under the NIHR Health Informatics Collaborative data sharing

agreement. Any request to access data can be made via imperial.cvhic@nhs.net referring to the title of this paper.

**Funding:** AK is funded by a British Heart Foundation clinical research training fellowship (FS/20/18/34972). AH is funded by a Wellcome Trust clinical research fellowship (WIII_P67144). ASVS is supported by a British Heart Foundation Intermediate Clinical Research Fellowship (FS/19/17/34172). ADS is funded by a THIS Institute postdoctoral fellowship. JACS is funded by National Institute for Health Research Senior Investigator award (NF-SI-0611-10168). AMS is funded by a British Heart Foundation Professorship (CH/1999001/11735). RSP is funded by a British Heart Foundation intermediate fellowship (FS/14/76/30933). JM is supported by the British Heart Foundation Imperial Centre for Research Excellence (RE/18/4/34215). RK is funded by a British Heart Foundation Intermediate Clinical Research Fellowship (FS/17/16/32560). The funders had no role in study design, data collection and analysis, decision to publish, or preparation of the manuscript.

**Competing interests:** I have read the journal's policy and the authors of this manuscript have the following competing interests: WK is a member of the Executive Steering Committee of CANTOS and has received modest amounts for consulting from Novartis). All remaining authors declare no competing interests.

**Abbreviations:** ACS, acute coronary syndrome; AUROC, area under the receiver operating characteristic curve; CAMI-1, CRP apheresis in Acute Myocardial Infarction; CANTOS, Canakinumab Anti-inflammatory Thrombosis Outcome Study; CIRT, Cardiovascular Inflammation Reduction Trial; COLCOT, Colchicine Cardiovascular Outcomes Trial; CRP, C-reactive protein; CV, cardiovascular; CVD, cardiovascular disease; HR, hazard ratio; hsCRP, high-sensitivity C-reactive protein; ICD, International Classification of Diseases; IDI, integrated discrimination improvement; IL, interleukin; IQR, interquartile range; LoDoCo2, Low Dose Colchicine; MACE, major adverse cardiovascular event; NHS, National Health Service; NRI, net reclassification index; NSTEMI, non-ST segment elevation myocardial infarction; PCI, percutaneous coronary intervention; SREC, Research Ethics Committee; STEMI, ST segment elevation myocardial infarction; STROBE, Strengthening the Reporting of Observational Studies in Epidemiology; ULN, upper limit of normal; WCC, white cell count.

2017. Patients were divided into 4 hsCRP groups (<2, 2 to 4.9, 5 to 9.9, and 10 to 15 mg/L). The main outcome measure was mortality within 3 years of index presentation. The association between hsCRP levels and all-cause mortality was assessed using multivariable Cox regression analysis adjusted for age, sex, haemoglobin, white cell count (WCC), platelet count, creatinine, and troponin.

Following the exclusion criteria, there were 102,337 patients included in the analysis (hsCRP <2 mg/L (*n* = 38,390), 2 to 4.9 mg/L (*n* = 27,397), 5 to 9.9 mg/L (*n* = 26,957), and 10 to 15 mg/L (*n* = 9,593)). On multivariable Cox regression analysis, there was a positive and graded relationship between hsCRP level and mortality at baseline, which remained at 3 years (hazard ratio (HR) (95% CI) of 1.32 (1.18 to 1.48) for those with hsCRP 2.0 to 4.9 mg/L and 1.40 (1.26 to 1.57) and 2.00 (1.75 to 2.28) for those with hsCRP 5 to 9.9 mg/L and 10 to 15 mg/L, respectively. This relationship was independent of troponin in all suspected ACS patients and was further verified in those who were confirmed to have an ACS diagnosis by clinical coding. The main limitation of our study is that we did not have data on underlying cause of death; however, the exclusion of those with abnormal WCC or hsCRP levels >15 mg/L makes it unlikely that sepsis was a major contributor.

## Conclusions

These multicentre, real-world data from a large cohort of patients with suspected ACS suggest that mildly elevated hsCRP (up to 15 mg/L) may be a clinically meaningful prognostic marker beyond troponin and point to its potential utility in selecting patients for novel treatments targeting inflammation.

## Trial registration

ClinicalTrials.gov - NCT03507309

---

## Author summary

### Why was this study done?

- C-reactive protein (CRP) is a protein produced by the liver and released into the bloodstream in response to inflammation. A blood test called a high-sensitivity C-reactive protein (hsCRP) accurately measures low levels of CRP.

- In patients presenting with a suspected heart attack, low levels of hsCRP on admission may prove useful in predicting death beyond using troponin, a marker of heart muscle damage.

### What did the researchers do and find?

- In this study of 102,337 patients with suspected heart attack, a higher hsCRP level was associated with a higher risk of death.

- This relationship was independent of the troponin level in all patients with suspected heart attack and was further verified in those who were confirmed to have a heart attack.

### What do these findings mean?

- These findings suggest that hsCRP is a clinically meaningful marker of risk of death in addition to troponin in patients with suspected heart attack.

- hsCRP has potential utility in selecting patients with heart attack for new treatments that target reducing inflammation.

## Introduction

Inflammation has been extensively studied and postulated as mechanistic in atherothrombosis [1]. Much debate has been on whether the immune system plays a protective or pathogenic role in atherogenesis and subsequent cardiovascular (CV) events. In addition to exploiting elements of the immune system as targets for novel therapies, some have been investigated for their role as potential biomarkers, aiming to permit better cardiovascular disease (CVD) risk stratification [2,3]. C-reactive protein (CRP) is the most widely evaluated of these biomarkers, given its relatively long circulating half-life and commonplace use in clinical practice.

In early observational studies of general and acute coronary syndrome (ACS) populations, high levels of CRP were found to represent a poorer prognosis, perhaps underlying a previously unrecognised inflammatory process relating to atherosclerosis [2,4–6]. Further studies have added significant weight to these observations, identifying that baseline serum CRP levels are an independent predictor of incident CV events and mortality in healthy individuals [7,8] as well as in patients with established CVD [9–12]. Although CRP has been shown to be a useful correlative biomarker in primary and secondary prevention, it is likely to be an end-protein readout of a mechanistic pathway involving interleukin (IL)-6, rather than a therapeutic target itself [13,14].

Considering high-sensitivity C-reactive protein (hsCRP) levels in a stable populations as a biomarker for selecting patients for advanced CV therapies in the modern era was pioneered by the Canakinumab Anti-inflammatory Thrombosis Outcome Study (CANTOS), a randomised, double-blind, placebo-controlled trial of canakinumab (a human monoclonal antibody targeted against IL-1β, upstream of IL-6), in preventing recurrent CV events. Patients in the canakinumab arm had a significant 15% reduction in CVD events over median 3.7 years of follow up, coexistent with reduced hsCRP levels and independent of lipid lowering [15]. Moreover, those that failed to achieve on-treatment hsCRP <2 mg/L with canakinumab saw no improvement in major adverse cardiac events (MACE) [16].

Results of the Cardiovascular Inflammation Reduction Trial (CIRT) had shed some doubt onto the role of modulating inflammation in atherosclerosis using rheumatological agents, after it terminated early and reported that low-dose methotrexate had no effect on CVD versus placebo [17]. However, this was not restricted to those with elevated hsCRP at baseline, suggesting that there may still be a significant treatment gap of focused inflammatory targeting in atherosclerosis. Somewhat surprisingly, no effects of methotrexate were seen on IL-6, IL-1β, or

hsCRP. More recent supportive evidence of targeting inflammation in CVD has been published, with the Colchicine Cardiovascular Outcomes Trial (COLCOT) [18] and the Low Dose Colchicine (LoDoCo2) for secondary prevention of CVD trial [19] reporting significant reductions in CV end points using colchicine, an immunomodulatory agent that works on several pathways including targeting IL-1β, in patients within 30 days of ACS or with chronic coronary disease. However, laboratory measures of inflammation (including CRP) at baseline from these studies are not available. If the baseline absolute risk of patients with ACS and elevated CRP values is high, then targeting these individuals with anti-inflammatory therapies will have larger clinical impact with lower values for the number needed to treat.

However, in patients presenting with suspected ACS, levels of hsCRP may either reflect background atherosclerotic disease activity or indeed also reflect the sequelae of the acute event. Nevertheless, hsCRP levels at presentation may prove useful in prognostication beyond routinely studied parameters, including troponin.

In this study, we aim to evaluate the prognostic value of mildly elevated hsCRP (up to 15 mg/L) beyond troponin in a large real-world cohort of unselected patients presenting with suspected ACS. Our hypothesis was that a mildly elevated hsCRP would be associated with mortality risk, beyond troponin, in patients with suspected ACS.

## Methods

We conducted a retrospective cohort study based on the National Institute for Health Research Health Informatics Collaborative data [20–22] of 257,948 patients with suspected ACS who had a troponin measured at 5 cardiac centres in the UK between 2010 and 2017 (Imperial College Healthcare, University College Hospital, Oxford University Hospital, Kings College Hospital and Guys and St Thomas" Hospital). Ethical approval for the dataset was obtained. The data acquisition and statistical analysis plan are available as S1 Data Aquisition and S1 Analysis Plan, respectively. There was no deviation in the statistical analysis plan during conduct of the study or following peer review. The study was registered at ClinicalTrials.gov, NCT03507309.

We developed a data model to capture the longitudinal record for patients with a suspected ACS, characterised by the request of a troponin test. For patients with multiple episodes of care for which troponin was tested, the first episode of care was used. We intentionally focused our population to those with normal white cell count (WCC) in an attempt to exclude overtly septic patients or those with haematological disorders that may have a major effect on levels of hsCRP. Furthermore, we excluded all patients with hsCRP >15 mg/L in a separate effort to exclude those with clinically significant inflammation or infection. All eligible patients were followed up, using routinely collected data on the National Health Service (NHS) Spine Application, until death or censoring in April 2017.

All analyses on hsCRP and other haematological and biochemical blood tests, apart from troponin, were performed using the first result measured during the hospital care episode.

The 99th percentile of the upper limit of normal (ULN) and the limit of detection of all hsCRP assays used across all cardiac centres were determined.

All analyses on troponin were performed using the peak troponin level. For patients who had a single troponin measurement, the peak troponin was based on this measurement. In the remainder of the patients who had more than one troponin test in the same hospital episode of care, the peak troponin value was defined as the highest of all measurements. We standardised the different troponin assays between the academic centres by scaling the results using the ratio of the observed troponin value divided by the ULN for each troponin assay. Each centre measured troponin I or troponin T using either contemporary or high-sensitivity assays (S1 Table). These tests yielded results in different measurable ranges, with unique cutoff points for the

ULN. There is current lack of standardisation of the various cardiac troponin assays as a physiological threshold for myocardial injury [23,24]. The ULN provided by the manufacturer for each troponin assay provides an analytical cutoff between normal and increased cardiac troponin levels. A positive troponin was defined as a result above the ULN for each troponin assay.

Patients who were admitted to the hospital had the International Classification of Diseases (ICD) discharge codes. Patients were classified as ACS based on the assigned ICD-10 codes (S2 Table) [25].

The population was divided into 4 categories, those with a normal hsCRP (<2 mg/L) [15] and 3 groups of incrementally increasing hsCRP (2 to 4.9 mg/L, 5 to 9.9 mg/L, and 10 to 15 mg/L). The groupings were selected to reflect low grade inflammation (overt infections were also excluded using the WCC data). The median baseline hsCRP in CANTOS was 4.2 mg/L; therefore, a smaller group of 2 to 4.9 mg/L was chosen to examine this cohort more closely [15]. The 4 hsCRP groups were further subdivided according to troponin positivity and subgroups including only patients diagnosed with an ACS during their index hospital admission.

This study was approved by the London South East Research Ethics Committee (REC reference: 16/HRA/3327). This study is reported as per the Strengthening the Reporting of Observational Studies in Epidemiology (STROBE) guideline (S1 Checklist).

## Statistical analysis

Descriptive statistics are displayed as median (interquartile range (IQR)) for continuous variables and number (%) for categorical variables. Comparisons between hsCRP groups with regard to baseline characteristics were explored using Kruskal–Wallis 1-way analysis of variance test and $\chi^2$ test for trend for continuous and categorical variables, respectively. The correlation between hsCRP and troponin concentrations was assessed using the Spearman correlation coefficient. The Kaplan–Meier method was used to calculate and display cumulative mortality, with hsCRP groups compared using the log-rank statistic. Subgroup analyses were performed for troponin positivity and ACS diagnosis.

Multivariable Cox regression analysis was applied to investigate whether the baseline hsCRP groups independently predict 3-year mortality after adjusting for other demographic and clinical variables. The proportional hazard assumption was violated with a significant relationship between the Schoenfield residuals of all covariates and time. Cox regression analysis with time-varying coefficients was therefore used with follow-up time divided into 6 intervals: <1 month, 1 to 3 months, 3 to 6 months, 6 to 12 months, 12 to 24 months, and 24 to 36 months. The proportional hazard assumption was met in each of these time intervals. The coefficients calculated during each of these time intervals were based on the variables measured during the index hospital admission. Furthermore, using Martingale residuals, nonlinearity was detected in the relationship between the log hazard and all continuous covariates. To model these nonlinear relationships, we used restricted cubic splines in the Cox proportional hazards model.

To assess the predictive role of hsCRP on short-term (30 days) and long-term (3 years) mortality beyond conventional risk factors and troponin, we established 3 statistical models. Model 1 was adjusted for age, sex, haemoglobin, and creatinine; model 2 was additionally adjusted for troponin positivity; model and 3 was further adjusted for hsCRP (hsCRP <2 mg/L, 2 to 4.9 mg/L, 5 to 9.9 mg/L, and 10 to 15 mg/L). The statistical models were based on the predicted probabilities of a logistic regression model. Ethnicity was not included in the risk model due to approximately 10% of patients having missing data. Furthermore, for those who had ethnicity recorded, approximately 15% had "other" recorded as an ethnic category, therefore making it less easy to interpret as part of a clinical risk model.

Discrimination of the different models was assessed by comparing the areas under the receiver operating characteristic curves (AUROCs) using the DeLong nonparametric approach. We also computed the continuous net reclassification index (NRI) and performed an integrated discrimination improvement (IDI) analysis with the censored survival data. Comparing the 3 models sequentially, the NRI reflects the net increment in prediction accuracy, while the IDI reflects the change in calculated risk for each individual. The IDI reflects the change in discrimination slope (i.e., difference between the mean estimated risk for patients who died and those who survived) of the model with the new predictor compared to the model with only the established predictors. IDI (range −100% to 100%) represents overall risk discrimination improvement.

In order to evaluate the performance of negative hsCRP and troponin values in predicting mortality, the negative predictive values of a negative hsCRP, negative troponin, or both a negative troponin and hsCRP were plotted against the range of hypothetical mortality rates at 30 days and 3 years. The negative predictive value, which varies with the prevalence of disease (mortality), is defined as the probability of not dying when the test result is negative.

Statistical analyses were performed using R 3.5.0 (R Core Team, Vienna, Austria) or MedCalc version 15.8 (MedCalc Software, Mariakerke, Belgium). The survIDINRI package on R was used to implement the IDI and NRI for comparing risk prediction models.

## Results

### Baseline population characteristics

A total of 257,948 patients presented with suspected ACS to 5 acute cardiac centres and were included in the NIHR Health Informatics Collaborative dataset. Moreover, 71,262 patients did not undergo hsCRP testing during their hospital care episode and were therefore excluded from further analysis. A further 67,206 patients were excluded based on hsCRP level >15 mg/L. The remaining 129,480 patients were divided into 4 groups by hsCRP levels, reflecting low grade inflammation. After further exclusion of patients with abnormal WCCs (<4 or >11 × $10^9$/L) or those who did not have a WCC blood test, 102,337 patients were included in the final analysis. The median follow-up was 3.3 years (IQR 1.4 to 5.1). The study cohort was divided into 4 hsCRP categories (hsCRP <2 mg/L ($n$ = 38,390), 2 to 4.9 mg/L ($n$ = 27,397), 5 to 9.9 mg/L ($n$ = 26,957), and 10 to 15 mg/L ($n$ = 9,593)). Overall, 53.3% ($n$ = 54,534) of patients in the study cohort were admitted to the hospital. Among these patients, 10.8% ($n$ = 5,910) were coded with an ACS during their inpatient stay. Patient flow through the study is shown in Fig 1. S3 Table displays the limits of detection and references ranges for the 99th percentile of the ULN for the hsCRP assays used in the cardiac centres included in the study. Of note, all assays had a lower limit of detection ≤0.2mg/L with assays validated internally by all institutions.

Table 1 displays the baseline clinical characteristics for the patients included in the analysis. Patients with higher hsCRP values were older and more likely to be female. Although there were significant differences in levels of troponin between hsCRP groups, there was no obvious trend. We tested this relationship further to establish if the variables were interdependent, finding a very weak correlation with Spearman coefficient R = 0.15 ($p$ < 0.001) (S1 Fig). Although we tested hsCRP in relation to "peak" troponin available in predicting mortality, we established that for patients with an elevated troponin result, the median number of days between the first admission hsCRP measured and peak troponin level was 3 (IQR 0 to 5) days, suggesting that the admission hsCRP represents presentation values rather than the sequelae of the index event.

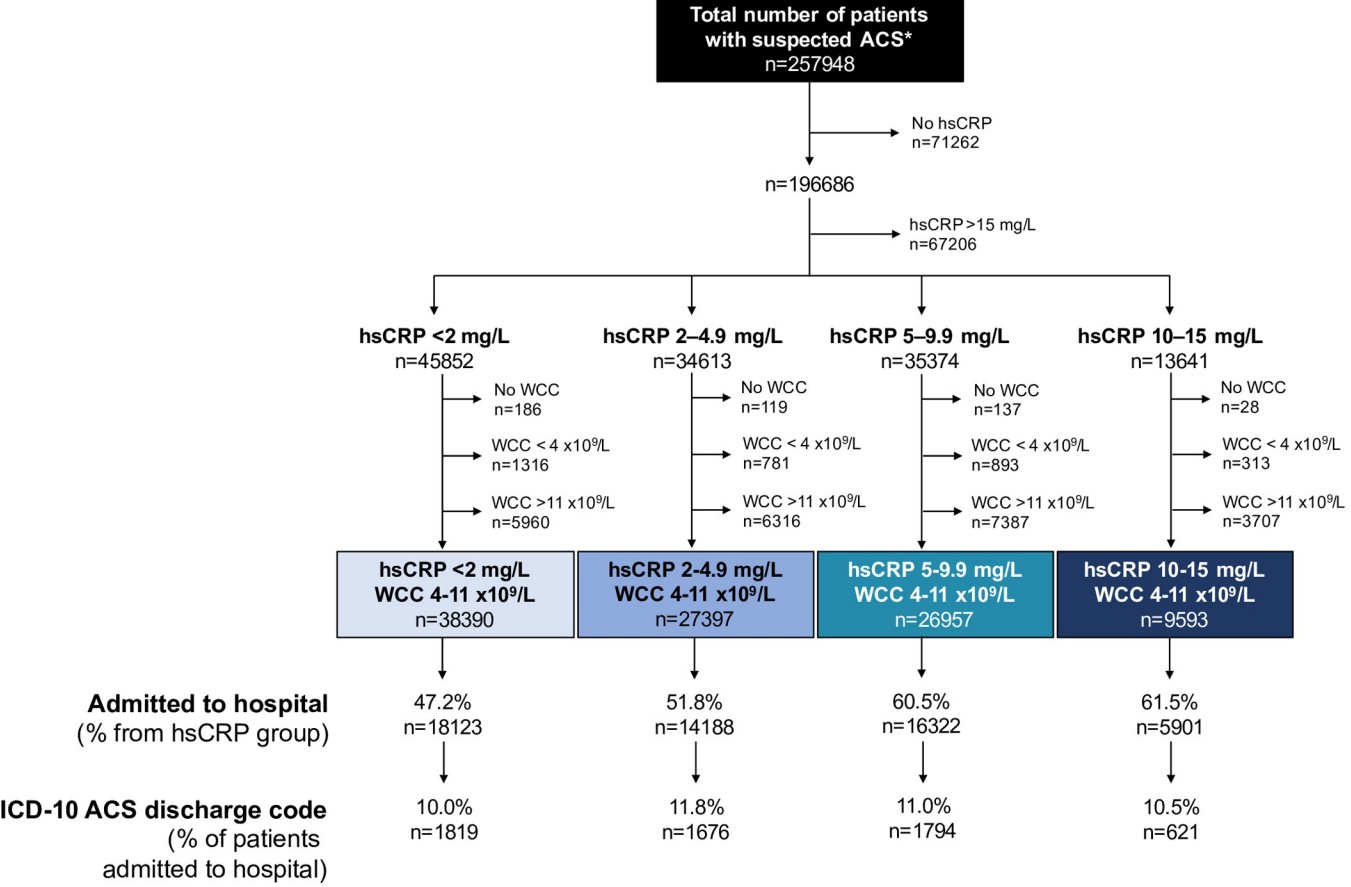

**Fig 1. Flow of patients through the study.** *Suspected ACS characterised by the request of a troponin. ACS, acute coronary syndrome; hsCRP, high-sensitivity C-reactive protein; ICD, International Classification of Diseases; WCC, white cell count.

## Higher hsCRP predicts significantly increased mortality risk

Patients in all 4 hsCRP categories had significantly incremental cumulative mortality over 3 years, in line with increasing hsCRP at baseline. Fig 2A displays cumulative mortality per hsCRP group, revealing increasing mortality with each consecutive group. Fig 2B further stratifies the groups according to dichotomised peak troponin level as positive or negative. This shows the greatest mortality for patients in the highest hsCRP group who also had a positive troponin assay (36.0% at 3 years). The additive effect of hsCRP to troponin on mortality is seen in all groups. There is also a graded mortality increase seen across the hsCRP troponin-negative groups, with lower mortality at 3 years than the troponin-positive groups. However, the troponin-negative highest hsCRP group (10 to 15 mg/L) had greater mortality at 3 years than the troponin-positive lowest hsCRP group (<2 mg/L) ($p < 0.001$). A similar relationship with hsCRP is seen in patients with different levels of troponin positivity (S2 Fig). We have Supporting information available on final ACS discharge diagnosis from clinical coding. Although these data have low granularity and are not available for all patients, we were able to use it to test if the relationship between hsCRP was still valid if we divided the population on discharge diagnosis of ACS, rather than suspected ACS on admission. Fig 2C demonstrates that this graded relationship between hsCRP and mortality was still present when restricting the cohort to just those clinically coded with confirmed ACS. A positive relationship between

**Table 1. Baseline clinical characteristics.**

| | hsCRP <2 mg/L (n = 38,390) | | hsCRP 2 to 4.9 mg/L (n = 27,397) | | hsCRP 5 to 9.9 mg/L (n = 26,957) | | hsCRP 10 to 15 mg/L (n = 9,593) | | p-Value |
|---|---|---|---|---|---|---|---|---|---|
| | n (median (IQR) or number (% of total patients)) | Total patients (% of all patients) | n (median (IQR) or number (% of total patients)) | Total patients (% of all patients) | n (median (IQR) or number (% of total patients)) | Total patients (% of all patients) | n (median (IQR) or number (% of total patients)) | Total patients (% of all patients) | |
| *General demographics* | | | | | | | | | |
| Age (years) | 59 (45 to 75) | 38,374 (99.96) | 65 (51 to 79) | 27,381 (99.94) | 64 (50 to 79) | 26,952 (99.98) | 70 (54 to 82) | 9,589 (99.96) | <0.001 |
| Male | 21,566 (56.20) | 38,374 (99.96) | 14,377 (52.51) | 27,378 (99.93) | 14,292 (53.03) | 26,951 (99.98) | 4,774 (49.79) | 9,589 (99.96) | <0.001 |
| Ethnicity | | | | | | | | | |
| White | 22,824 (71.04) | 32,130 (83.69) | 17,157 (73.69) | 23,284 (84.99) | 16,698 (70.54) | 23,671 (87.81) | 6,095 (74.37) | 8,195 (85.43) | <0.001 |
| South Asian | 1,445 (4.50) | | 993 (4.26) | | 879 (3.71) | | 304 (3.71) | | |
| Black | 3,018 (9.39) | | 2,197 (9.44) | | 3,367 (14.22) | | 960 (11.71) | | |
| Other | 4,843 (15.07) | | 2,937 (12.61) | | 2,727 (11.52) | | 836 (10.20) | | |
| *Haematology/biochemistry* | | | | | | | | | |
| Haemoglobin (g/dL) | 13.8 (12.7 to 14.9) | 38,389 (100) | 13.6 (12.4 to 14.7) | 27,395 (99.99) | 13.4 (12.1 to 14.5) | 26,954 (99.99) | 13.0 (11.7 to 14.2) | 9,592 (99.99) | <0.001 |
| WCC (×10⁹/L) | 7.1 (5.9 to 8.5) | 38,390 (100) | 7.5 (6.2 to 8.9) | 27,397 (100) | 7.6 (6.3 to 9.0) | 26,957 (100) | 7.9 (6.5 to 9.3) | 9,593 (100) | <0.001 |
| Platelet count (×10⁹/L) | 225 (190 to 264) | 38,373 (99.96) | 230 (192 to 272) | 27,391 (99.98) | 226 (185 to 270) | 26,941 (99.94) | 230 (186 to 281) | 9,589 (99.96) | <0.001 |
| Creatinine (μmol/L) | 74 (64 to 88) | 38,279 (99.71) | 76 (65 to 92) | 27,338 (99.78) | 76 (64 to 94) | 26,882 (99.72) | 78 (65 to 100) | 9,584 (99.91) | <0.001 |
| Positive troponin | 6,099 (15.9) | 38,390 (100) | 5,937 (21.7) | 27,397 (100) | 7,402 (27.5) | 26,957 (100) | 3,126 (32.6) | 9,593 (100) | <0.001 |
| Troponin level (×ULN) | 0.003 (0.003 to 0.4) | 38,390 (100) | 0.003 (0.003 to 0.7) | 27,397 (100) | 0.23 (0.003 to 1.25) | 26,957 (100) | 0.06 (0.003 to 1.7) | 9,593 (100) | <0.001 |

Note: The table uses $\times 10^9/L$ and $\times$ULN notation as shown.

p-Values were calculated using Kruskal–Wallis 1-way analysis of variance and $\chi^2$ test for trend for continuous and categorical variables, respectively.

hsCRP, high-sensitivity C-reactive protein; IQR, interquartile range; ULN, 99th percentile of the upper limit of normal; WCC, white cell count.

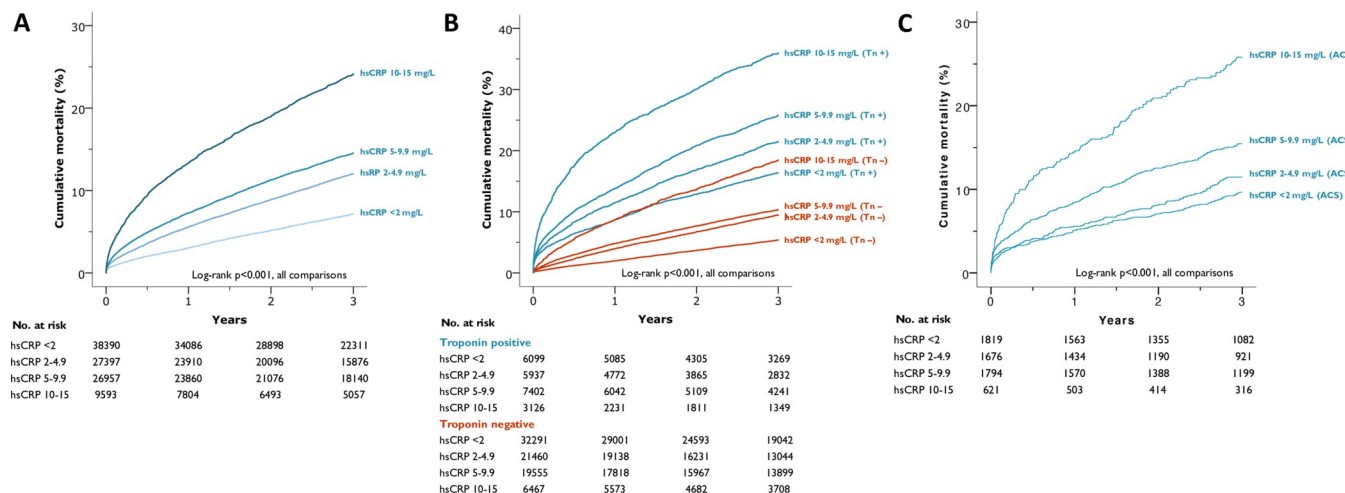

**Fig 2. Unadjusted Kaplan–Meier mortality curves by (A) hsCRP level, (B) hsCRP level and troponin positivity, and (C) hsCRP level in a subgroup of patients with ACS.** ACS, acute coronary syndrome; hsCRP, high-sensitivity C-reactive protein; Tn +, troponin positive, Tn −, troponin negative.

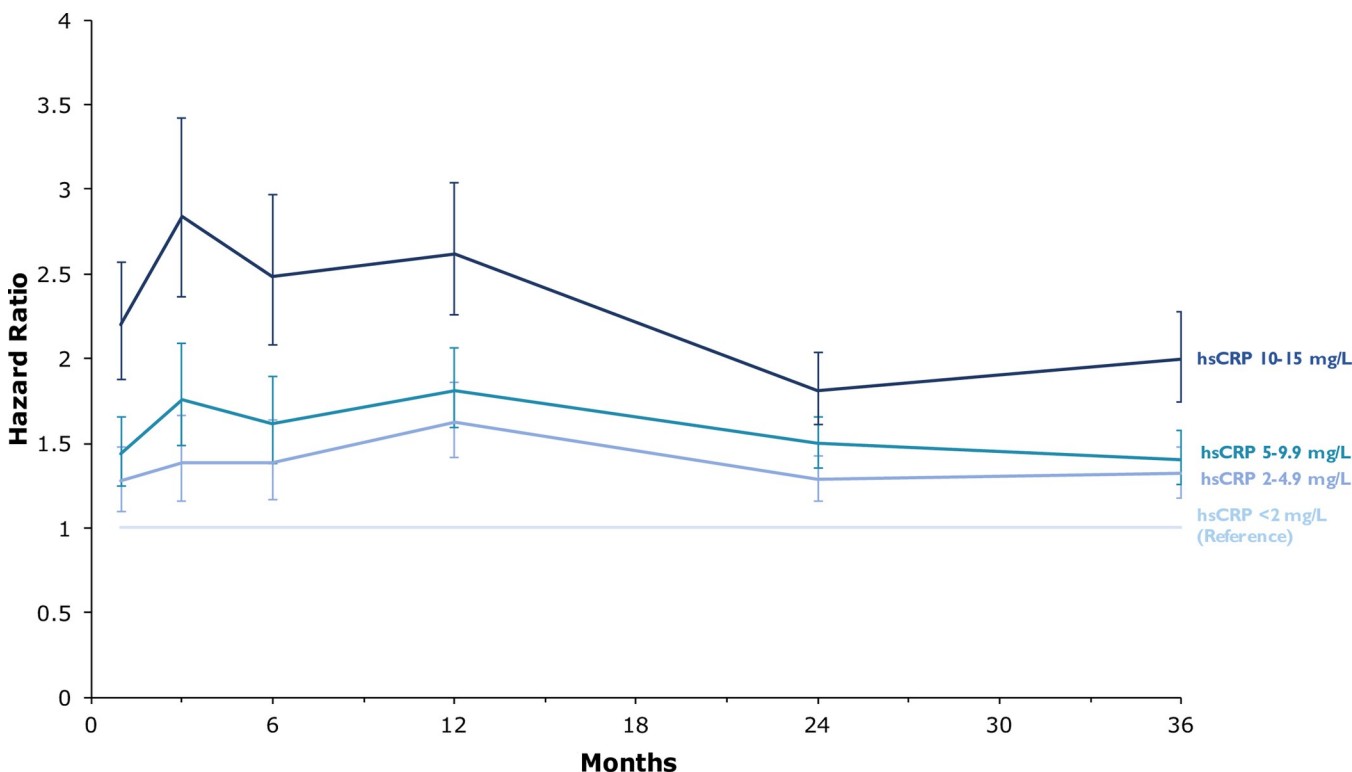

**Fig 3. Multivariable Cox regression analysis with time-varying coefficients.** HRs are adjusted for age, sex, haemoglobin, WCC, platelet count, creatinine, and troponin level. The adjusted HRs were calculated during the following time periods: <1 month, 1 to 3 months, 3 to 6 months, 6 to 12 months, 12 to 24 months, and 24 to 36 months. The HRs are displayed at the end of each respective time period. The error bars depict 95% CIs. HR, hazard ratio; hsCRP, high-sensitivity C-reactive protein; WCC, white cell count.

hsCRP level and mortality was also observed in patients admitted to the hospital without a clinically coded diagnosis of ACS (S3 Fig).

When undertaking Cox regression analysis with time-varying coefficients, hsCRP, albeit mildly raised, was an independent predictor of mortality over time, after adjusting for available clinically relevant covariates: age, sex, haemoglobin, WCC, platelet count, creatinine, and troponin positivity (Fig 3). At 30-day follow-up, the effect of even a very modestly raised hsCRP is evident, demonstrating a graded relationship with mortality (hazard ratio (HR) 1.27 (95% CI 1·10 to 1·48), HR 1.44 (95% CI 1.25 to 1.66), and HR 2.20 (95% CI 1.88 to 2.57) for hsCRP groups 2 to 4.9 mg/L, 5 to 9.9 mg/L, and 10 to 15 mg/L respectively, relative to a normal CRP level (<2 mg/L)). This effect persisted up to 3 years in the study, with HR (95% CI) of 1.32 (1.18 to 1.48) for those with hsCRP 2.0 to 4.9 mg/L, 1.40 (1.26 to 1.57), and 2.00 (1.75 to 2.28) for those with hsCRP 5 to 9.9 mg/L and 10 to 15 mg/L, respectively. The HRs for mortality risk at 3 years in the hsCRP groups stratified by troponin positivity are shown in S4 Table.

## hsCRP significantly improves mortality risk prediction

We next explored whether inclusion of hsCRP could better reclassify the population into at-risk mortality groups. The association with short-term and long-term mortality was assessed using 3 different risk models (model 1: age, sex, haemoglobin, and creatinine; model 2: model 1 plus troponin (positivity versus negativity); and model 3: model 2 plus hsCRP groups (<2, 2 to 4.9, 5 to 9.9, and 10 to 15 mg/L) (Table 2). For cumulative mortality at each time point, each successive model was better able to discriminate risk than its precursor ($p < 0.001$), such that

**Table 2. hsCRP risk model discrimination, calibration, and reclassification.**

| | 30-day mortality | | | 3-year mortality | | |
|---|---|---|---|---|---|---|
| | **Model 1** | **Model 2** | **Model 3** | **Model 1** | **Model 2** | **Model 3** |
| **Basic demographic/ biochemistry** | Age, sex, haemoglobin, and creatinine | Age, sex, haemoglobin, and creatinine | Age, sex, haemoglobin, and creatinine | Age, sex, haemoglobin, and creatinine | Age, sex, haemoglobin, and creatinine | Age, sex, haemoglobin, and creatinine |
| **+ Troponin** | | + Troponin (positive versus negative) | + Troponin (positive versus negative) | | + Troponin (positive versus negative) | + Troponin (positive versus negative) |
| **+ hsCRP** | | | + hsCRP groups (<2, 2 to 4.9, 5 to 9.9, and 10 to 15) | | | + hsCRP groups (<2, 2 to 4.9, 5 to 9.9, and 10 to 15) |
| *Discrimination* | | | | | | |
| AUROC | 0.747 | 0.806 | 0.812 | 0.790 | 0.797 | 0.803 |
| 95% CI | 0.745 to 0.750 | 0.803 to 0.808 | 0.810 to 0.815 | 0.787 to 0.793 | 0.794 to 0.800 | 0.800 to 0.806 |
| *p*-Value | <0.001 | <0.001 | <0.001 | <0.001 | <0.001 | <0.001 |
| *p*-Value (versus Model 1) | - | <0.001 | <0.001 | - | <0.001 | <0.001 |
| *p*-Value (versus Model 2) | - | - | <0.001 | - | - | <0.001 |
| *Reclassification* | | | | | | |
| | | versus Model 1 | versus Model 2 | | versus Model 1 | versus Model 2 |
| IDI (%) | - | 1.1 | 0.3 | - | 1.0 | 0.9 |
| 95% CI | - | 0.9 to 1.3 | 0.2 to 0.4 | - | 0.8 to 1.2 | 0.7 to 1.1 |
| *p*-Value | - | <0.001 | <0.001 | - | <0.001 | <0.001 |
| NRI (%) | - | 42.6 | 18.4 | - | 20.9 | 13.9 |
| 95% CI | - | 39.7 to 45.3 | 14.6 to 21.0 | - | 19.2 to 22.0 | 12.7 to 14.9 |
| *p*-Value | - | <0.001 | <0.001 | - | <0.001 | <0.001 |

Discrimination of the different models was assessed by comparing the AUROC using the DeLong nonparametric approach. The survIDINRI package on R was used to implement the IDI and NRI for comparing risk prediction models. Likelihood ratio test *p*-values have been calculated, which confirm the significance of the predictive performance testing using IDI and NRI. The *p*-values remained significant at <0.001 for all model comparisons.

AUROC, area under the receiver operating characteristic curve; hsCRP, high-sensitivity C-reactive protein; IDI, integrated discrimination improvement; NRI, net reclassification index.

inclusion of troponin and hsCRP gave the most robust risk discrimination. Of note, model 3, which includes both troponin positivity and hsCRP categories, achieves an AUROC >0.8 at 30 days and 3-year mortality, surpassing the use of troponin on its own. The addition of model 3 over model 2 resulted in an IDI of 0.3% and 0.9% at 30 days and 3 years, respectively (all $p < 0.001$), as well as the highest overall NRI of 18.4% and 13.9% at the same time points, respectively (all $p < 0.001$). Thus, the inclusion of hsCRP to the basic model with troponin level significantly improved all-cause mortality risk prediction.

Fig 4 displays the relationship between the negative predictive value of hsCRP and troponin testing with hypothetical mortality. For a potential rule-out test based on risk prediction, there was an improvement in negative predictive values for the lowest hsCRP group (<2 mg/L) in addition to troponin negativity, compared to either test being negative alone, at 30 days and 3 years.

## Discussion

In this large retrospective cohort study, we demonstrate a positive and graded relationship between mildly elevated hsCRP levels (up to 15 mg/L) and all-cause mortality over 3 years.

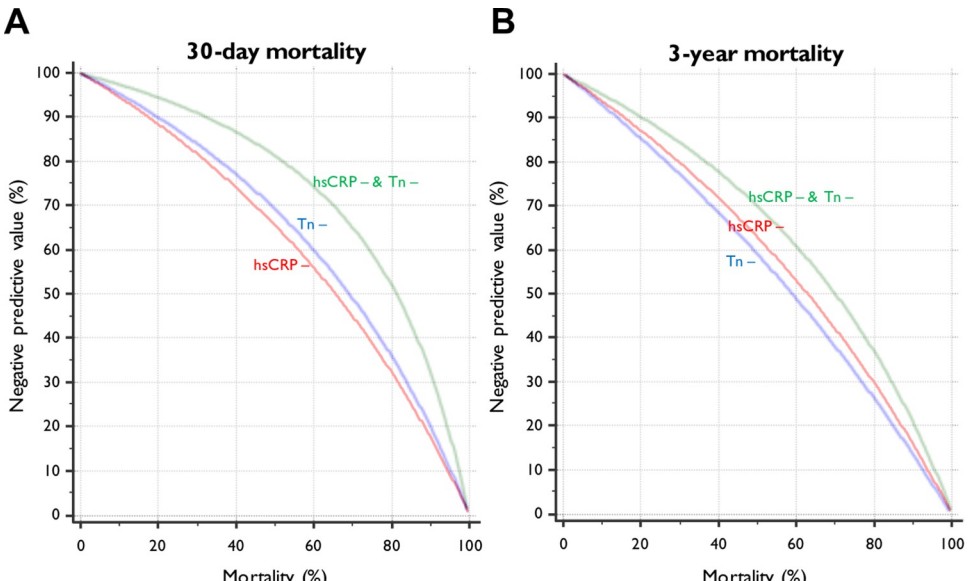

**Fig 4. Relationship between negative predictive value of hsCRP and troponin testing with (A) 30-day and (B) 3-year mortality.** hsCRP, high-sensitivity C-reactive protein; hsCRP −, negative high-sensitivity C-reactive protein; Tn −, negative troponin.

The mortality risk associated with increased hsCRP was independent of age, sex, haemoglobin, WCC, platelet count, creatinine, and troponin positivity, suggesting its value as a clinically useful biomarker in the context of suspected ACS. Compared to a normal hsCRP (<2 mg/L), a mildly raised hsCRP between 10 and 15 mg/L was associated with a 2.2-fold increased mortality risk at 30 days, which persisted at 3 years. In this study, patients who had negative serum troponin but a hsCRP between 10 and 15 mg/L had equivalent mortality to those who were troponin positive but had a normal hsCRP, <2 mg/dL, in Kaplan–Meier analysis at 3 years. Moreover, patients were better reclassified into at-risk mortality groups when hsCRP was included in addition to troponin, suggesting a potential use in this clinical setting.

We limited the cohort by prespecifying strict exclusion criteria that aim to reduce the numbers of patients with infective, inflammatory, or malignant disorders. Therefore, we excluded anyone with abnormal WCC or hsCRP >15 mg/L, with the aim of assembling a cohort that would be informative for clinicians treating patients with suspected ACS. This was possible due to the large population available for study.

The key finding of this study was that within a group of patients with low levels of hsCRP elevation, there was a significant increase in mortality risk in patients presenting with suspected ACS in both the short and long term. In Cox regression analysis with time-dependent covariates, the highest hsCRP group had a HR 2.20 (95% CI 1.88 to 2.57) at baseline, which only marginally attenuated to 2.00 (95% CI 1.75 to 2.28) at 3 years. This long-term risk is clinically significant and may present an opportunity to improve patient selection for novel diagnostics and therapeutics. The striking finding that patients in the highest hsCRP group, but who were troponin negative, had similar mortality to those who were troponin positive but had hsCRP < 2 mg/L, brings into question whether, in clinical practice, we are missing high-risk patients who may benefit from further investigation and treatment [10]. To expand on this, we believe that it is common for patients with a low-normal hsCRP level to be discharged from the emergency department with a caution that they may have a viral infection or nonspecific inflammation, with no comment on mortality risk stratification. Reliance on troponin for

risk stratification alone may mean that signals suggesting patient vulnerability are not followed up appropriately. Importantly, the relationships seen with hsCRP and mortality hold when the population is restricted to those with a diagnosis of ACS, rather than suspected ACS.

There has been a very positive response surrounding immunomodulatory drugs in CVD following publication of the CANTOS [15], COLCOT [18], and LoDoCo2 trials [19]. However, CIRT [17] highlighted that not all anti-inflammatory agents would show promise in treating atherosclerosis. Methotrexate in this study had hardly any effect on hsCRP levels, and in tandem, did not reduce CV events, warranting early study termination. Unfortunately, no CRP data from COLCOT or LoDoCo2 are available. Of note, a recent Phase II trial (RESCUE) assessed the efficacy of ziltivekimab, a novel monoclonal antibody against the IL-6 ligand, in patients with hsCRP >2mg/L and chronic kidney disease. This study demonstrated a dose-related reduction in hsCRP (92% with the highest dose), and a large CV outcomes trial is planned [26].

It is difficult in clinical practice to identify the exact time point when an ACS event occurs. The time of diagnosis may not coincide with the time of onset of the ACS event. In order to determine the effect of baseline hsCRP levels rather than hsCRP rises in response to an infarct, we carried out analyses on admission hsCRP levels in comparison to the "peak" troponin level in predicting mortality. For patients with an elevated troponin result, the median number of days between the first admission hsCRP measured and peak troponin level was 3 (IQR 0 to 5) days, infering that the admission hsCRP levels likely represent presentation values rather than the sequelae of the index event.

We excluded individuals who had a CRP >15 mg/L (or abnormal WCC) to limit the population to those without overt infections, malignancies, or systemic inflammatory conditions. It is unlikely that we excluded many patients with ACS due to the low CRP levels measured in patients with ACS in previous studies. In a recent study investigating the association of initial hsCRP levels with MACE and mortality after ACS [27], the median baseline hsCRP level was 10.5 mg/L (IQR, 4.2 to 30.3 mg/L) in ACS patients. On multivariable analysis, higher baseline hsCRP level (HR, 1.36 [95% CI, 1.13 to 1.63]; $p = 0.001$) was also shown to be independently associated with MACE and mortality.

Recently, in the multicentre matched-control pilot study on CRP apheresis in Acute Myocardial Infarction (CAMI-1), there was a correlation between the CRP gradient between 12 and 32 hours after symptom onset and both myocardial infarct size and the extent of reduction in left ventricular function [28]. The CAMI-1 investigators evaluated the feasibility and safety of CRP apheresis to reduce myocardial infarct size after ST segment elevation myocardial infarction (STEMI) as determined by CV magnetic resonance. While the primary end point of reduction of infarct size was not met, a sufficiently large randomised trial of CRP apheresis therapy in uncomplicated STEMI with the end point of infarct size reduction is currently under development at the Medical University of Innsbruck. This should provide further insight as to whether CRP is also a potentially damaging mediator for CV disease, infarct size, and subsequent mortality risk [29]. However, these results and planned trials need to be viewed in the context of the large mendelian randomisation meta-analysis that concluded that CRP itself [13] (unlike the the IL-6 receptor [30]) is not causally related to coronary heart disease. Indeed, the data from the present study may influence future trial design and lead to careful consideration of the use of hsCRP to select patients at higher CV risk who may derive greater benefit from targeted treatment.

Moreover, an interesting finding in the present study is that those with raised hsCRP and moderate troponin elevations had the greatest mortality risk. While there was a graded relationship between hsCRP level and mortality among patients with different levels of troponin positivity, the mortality risk in the hsCRP groups increased progressively up to >5 to 10×ULN

of troponin and then progressively declined in the higher troponin groups. Previous research from our group has demonstrated this inverted U-shaped relation between troponin level and mortality in an unselected cohort of patients undergoing troponin testing. This was primarily due to an interaction between ACS diagnosis and differential receipt of invasive management in this cohort. [21] The >5 to 10×ULN troponin group are likely to represent non-ST segment elevation myocardial infarctions (NSTEMI) than STEMI, and it is perhaps in this population that any targeted anti-inflammatory therapy would have the most clinical benefit.

The observed discrimination and net reclassification when adding hsCRP to our risk models is an important new finding. There have been several studies showing the value of hsCRP as an added covariate for predicting long-term mortality risk in primary prevention populations or those with chronic CVD, leading to developments such as the Reynolds risk score, which improves on the widely adopted Framingham risk score [31]. To our knowledge, there have been no studies that have demonstrated significantly superior AUROC in concert with changes in IDI and NRI that are clinically relevant.

There are some limitations to this study. The present study had access to all-cause, but not specific, mortality data. This limits our ability to understand the causes of death in the patients with the higher hsCRP levels. However, the exclusion of those with abnormal WCC or hsCRP levels >15 mg/L makes it unlikely that sepsis was a major contributor to the excess mortality. Nonetheless, all-cause mortality is clearly a clinically relevant end point. A further limitation is the retrospective nature of the data, where covariates were collated from clinical records, rather than a prospective database that is designed to answer a predefined hypothesis. We were unable to determine the patient comorbidities for our cohort. The median age demographic of patients progressively increases from the lowest to the highest quartile of hsCRP. While we were unable to adjust our analyses by patient comorbidities, all survival analyses were adjusted for age, which may also act as a surrogate marker of patient comorbidities, which are expected to increase in frequency with older age.

The data from this study emphasises the need for further prospective work that explores the inflammation pathway as a therapeutic target to address the still unanswered mortality gap in patients presenting with suspected ACS. We know that recurrent MACE in those undergoing percutaneous coronary intervention (PCI) [32] or presenting with ACS is still unacceptably elevated and shown to be as high as 20% at 3 years [33] despite optimal medical therapy. Our findings indicate a means of selecting those high-risk patients that may benefit from novel therapeutics.

## Conclusions

The data from this large real-world population in a modern healthcare system emphasise the importance of hsCRP in this setting and warrant attention and consideration from clinical guideline committees to include hsCRP in risk stratification of patients presenting with suspected ACS.

## Disclaimers

The views expressed in this publication are those of the authors and not necessarily those of the NHS, the NIHR or the Department of Health.

## Supporting information

**S1 Checklist. STROBE guideline checklist.** STROBE, Strengthening the Reporting of Observational Studies in Epidemiology.
(DOCX)

**S1 Data Acquisition. Data collection and access plan.**
(DOCX)

**S1 Analysis Plan. Statistical analysis plan.**
(DOCX)

**S1 Table. Troponin assays at participating cardiac centres.**
(DOCX)

**S2 Table. ICD-10 diagnostic codes used to indicate an ACS diagnosis.** ACS, acute coronary syndrome; ICD, International Classification of Diseases.
(DOCX)

**S3 Table. hsCRP assays at participating cardiac centres.** hsCRP, high-sensitivity C-reactive protein.
(DOCX)

**S4 Table. Adjusted HRs for 3-year mortality according to troponin and hsCRP stratified groups.** HR, hazard ratio; hsCRP, high-sensitivity C-reactive protein.
(DOCX)

**S1 Fig. Correlation between hsCRP and troponin level.** hsCRP, high-sensitivity C-reactive protein.
(DOCX)

**S2 Fig. Kaplan–Meier mortality curve by hsCRP level and different levels of troponin positivity.** hsCRP, high-sensitivity C-reactive protein.
(DOCX)

**S3 Fig. Unadjusted Kaplan–Meier mortality curves by ACS diagnosis.** ACS, acute coronary syndrome.
(DOCX)

## Acknowledgments

This paper reports independent research led and funded by the National Institute for Health Research (NIHR) Imperial Biomedical Research Centre (BRC), as part of the NIHR Health Informatics Collaborative with NIHR Oxford BRC, NIHR University College London Hospitals BRC, NIHR Guy's & St Thomas' BRC, NIHR Cambridge BRC, NIHR Bristol BRC, NIHR Birmingham BRC, NIHR Leeds BRC, NIHR Leicester BRC, NIHR Manchester BRC, NIHR Royal Marsden BRC, and NIHR Southampton BRC.

## Author Contributions

**Conceptualization:** Amit Kaura, Adam Hartley, Jamil Mayet, Ramzi Khamis.

**Data curation:** Amit Kaura, Ben Glampson, Abdulrahim Mulla.

**Formal analysis:** Amit Kaura.

**Funding acquisition:** Amit Kaura, Jamil Mayet.

**Investigation:** Amit Kaura, Adam Hartley, Vasileios Panoulas, Ben Glampson, Anoop S. V. Shah, Jim Davies, Abdulrahim Mulla, Kerrie Woods, Joe Omigie, Anoop D. Shah, Mark R. Thursz, Paul Elliott, Harry Hemmingway, Bryan Williams, Folkert W. Asselbergs, Michael

O'Sullivan, Graham M. Lord, Adam Trickey, Jonathan AC Sterne, Dorian O. Haskard, Narbeh Melikian, Darrel P. Francis, Wolfgang Koenig, Ajay M. Shah, Rajesh Kharbanda, Divaka Perera, Riyaz S. Patel, Keith M. Channon, Jamil Mayet, Ramzi Khamis.

**Methodology:** Amit Kaura, Adam Hartley, Anoop S. V. Shah, Ramzi Khamis.

**Project administration:** Jamil Mayet, Ramzi Khamis.

**Resources:** Ben Glampson.

**Software:** Ben Glampson.

**Supervision:** Jamil Mayet, Ramzi Khamis.

**Validation:** Anoop S. V. Shah.

**Visualization:** Amit Kaura, Ramzi Khamis.

**Writing – original draft:** Amit Kaura, Adam Hartley.

**Writing – review & editing:** Amit Kaura, Adam Hartley, Dorian O. Haskard, Wolfgang Koenig, Jamil Mayet, Ramzi Khamis.

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
