## [Editor Report · Decision Letter 0]

16 Sep 2021

Dear Dr Hartley, 

Thank you for submitting your manuscript entitled "High sensitivity C-reactive protein predicts mortality beyond troponin in 102337 patients with suspected acute coronary syndrome" for consideration by PLOS Medicine.

Your manuscript has now been evaluated by the PLOS Medicine editorial staff as well as by an academic editor with relevant expertise and I am writing to let you know that we would like to send your submission out for external peer review.

Please re-submit your manuscript within two working days, i.e. by Sep 20 2021 11:59PM.

Kind regards,

Callam Davidson

Senior Editor

PLOS Medicine

---

## [Decision Letter · Decision Letter 1]

4 Nov 2021

Dear Dr. Hartley,

Thank you very much for submitting your manuscript "High sensitivity C-reactive protein predicts mortality beyond troponin in 102337 patients with suspected acute coronary syndrome" (PMEDICINE-D-21-03948R1) for consideration at PLOS Medicine. 

Your paper was evaluated by an associate editor and discussed among all the editors here. It was also discussed with an academic editor with relevant expertise, and sent to independent reviewers, including a statistical reviewer. The reviews are appended at the bottom of this email and any accompanying reviewer attachments can be seen via the link below:

[LINK]

In light of these reviews, I am afraid that we will not be able to accept the manuscript for publication in the journal in its current form, but we would like to consider a revised version that addresses the reviewers' and editors' comments. Obviously we cannot make any decision about publication until we have seen the revised manuscript and your response, and we plan to seek re-review by one or more of the reviewers. 

We expect to receive your revised manuscript by Nov 25 2021 11:59PM. Please email us (plosmedicine@plos.org) if you have any questions or concerns.

We look forward to receiving your revised manuscript. 

Sincerely,

Callam Davidson, 

PLOS Medicine

plosmedicine.org

Please revise your title according to PLOS Medicine's style. Your title must be nondeclarative and not a question. It should begin with main concept if possible. "Effect of" should be used only if causality can be inferred, i.e., for an RCT. Please place the study design ("A randomized controlled trial," "A retrospective study," "A modelling study," etc.) in the subtitle (ie, after a colon).

Please structure your abstract using the PLOS Medicine headings (Background, Methods and Findings, Conclusions).

Abstract Background: Provide the context of why the study is important. The final sentence should clearly state the study question.

Abstract Methods and Findings:

* Please include the important dependent variables that are adjusted for in the analyses.

Financial disclosure: your financial disclosure does not detail the funding source, please provide this information in your response to the submission form. The detailed funding information at the end of the main text should be removed and provided instead as part of your response to the submission form (this will then be published as metadata in the event of publication). 

Similar to the above, please remove the ‘Authors’ Contributions’ and ‘Conflict of Interest’ sections from the end of the main text and ensure this information is captured in your submission form (this is essential as your submission form currently states no competing interests exist which is at odds with the main text).

You state in your Data Availability Statement that some restrictions will apply but also then state that all data is contained within the manuscript and/or supporting information files – please update for consistency, bearing in mind our data availability requirements (https://journals.plos.org/plosmedicine/s/data-availability).

Please update your manuscript to include line numbering in the margin.

Please update your citations to be in square brackets. 

Please relocate the information pertaining to the ethical approval from the end of the main text to the methods section.

I could not locate your data acquisition and statistical analysis plan? Please ensure these are included and, when stating that the data acquisition and statistical analysis plan are available in the supplementary material, please reference the relevant section of the supplementary material (e.g. S1).

Please only report P values to 3 decimal places (P<0.001 rather than P<0.0001). 

Your study is observational and therefore causality cannot be inferred. Please remove language that implies causality, such as ‘confirms its value as a biomarker’. Refer to associations instead or tone down the language used (e.g. suggests, potential, etc.).

Please add ‘to our knowledge’ when stating that ‘there have been no studies that have demonstrated significantly superior AUROC’ in paragraph 5 of your discussion.

Please remove the One-sentence summary.

References: journal name abbreviations should be those found in the National Center for Biotechnology Information (NCBI) databases.

Please ensure that the study is reported according to the STROBE guideline, and include the completed STROBE checklist as Supporting Information. Please add the following statement, or similar, to the Methods: "This study is reported as per the Strengthening the Reporting of Observational Studies in Epidemiology (STROBE) guideline (S1 Checklist)."

Comments from the reviewers:

Reviewer #1: "High sensitivity C-reactive protein predicts mortality beyond troponin in 102337 patients with suspected acute coronary syndrome" studies the association of (mildly) elevated levels of high-sensitivity C-reactive protein (hsCRP) on long-term (3 year) mortality risk in patients with suspected acute coronary syndromes (ACS), independent of troponin level. The main analysis concluded that there was indeed a positive and graded relationship between hsCRP and mortality at baseline and over three years, as summarized in Figures 2B and 3. In particular, stratification by hsCRP level demonstrated increasing 3-year mortality risk given fixed troponin positivity/negativity status (Figure 2B), and increased mortality over the full 3-year period by multivariable time-dependent Cox regression mortality analysis (Figure 3). hsCRP was therefore claimed to be a possible clinically prognostic marker in addition to troponin.

While the primary association between hsCRP and mortality risk appears clear, a number of issues might be considered, largely towards clarifying the methodology:

1. The definition of the upper limit of normal (ULN) for troponin assays might be explicitly stated, and the significance of "99th percentile of ULN" explained, especially since levels of troponin >100x ULN appear possible from Supplementary Figure 2. In particular, if ULN is the threshold on troponin level that separates troponin positive and negative diagnoses, then it would appear more natural to refer to the "99th percentile of troponin values" (to eliminate outliers at the upper end?); this might not be obvious to the more general reader.

2. Moreover, it is stated that the different troponin assays from various academic centres were standardized by scaling results using the ratio of observed troponin value divided by the (unique) ULN for each troponin assay. It might be clarified as to how the ULN for each troponin assay was determined. Is this ULN inherent to the troponin assay model, or empirically calculated on some cohort? Also, were such (scaling) normalizations significant?

3. Supplementary Figure 2 shows sensitivity analyses comparing the mortality curves for different definitions of troponin positivity (1-5x ULN, 5-10x, 10-100x, >100x). It appears that overall mortality risk is generally highest for 5-10x ULN, and actually lower for >100x ULN, than 1-5x ULN, i.e. the association of troponin level with mortality risk is nonlinear (as also briefly raised as an interesting point in the Discussion section). If so, it might be discussed in greater detail as to whether this nonlinear association is expected, and its possible causes.

4. It is stated that the data model involves patients with suspected ACS, as characterized by the request of a troponin test. It might be clarified as to whether this set of patients with troponin test results corresponds to the n=257,948 patients of the initial cohort (Figure 1), and whether the troponin test is exclusively administered towards suspected ACS.

5. It is stated that n=134,517 patients were admitted to hospital and thus had ICD discharge codes, and patients were classified as ACS based on said ICD-10 codes. It might be clarified as to how and where this n=134,517 can be represented in the patient flowchart (Figure 1)

6. It is stated that three statistical risk models were developed for mortality risk prediction (Table 2). The details of these models might be clarified - i.e. are they linear/logistic regression models (and if so, their coefficients might be provided), or other machine learning models, etc. Also, the choice of variables to match for/adjust for (e.g. ethnicity not included) might be briefly justified.

7. For the Kaplan-Meier analysis in Figure 2, it might be explicitly stated if any matching/adjustment was attempted. The "No. at risk" matrices might also be briefly explained/defined.

8. For Figure 3, it might be considered to provide additional time-dependent Cox regression mortality analysis charts stratified on troponin positivity/negativity, for consistency with the previous Kaplan-Meier analysis in Figure 2.

9. Further on Figure 3, the time-dependent analysis would appear to require data from patients at each follow-up interval over the full 3-year period, whereas there does not appear to be a description for the treatment of patients with missing/incomplete data, in Figure 1 and elsewhere. It might be clarified whether all required data was available for all patients involved, or if not, how missing data was handled (this extends to other covariates, other than hsCRP)

10. Figure 4 might be explained in greater detail, with "negative predictive value" in particular being clarified.

11. In the Results section, it is stated that "...Although we tested hsCRP in relation to peak troponin available in predicting mortality, we established that the median number of days between first troponin test and hsCRP was 0 (IQR 0), indicating that our hsCRP and troponin values represent presentation values rather than the sequelae of the index event". The significance of this statement might be clarified further. Does it imply that peak troponin is generally attained on the first test?

12. The claim of "...The independence of hsCRP from all other available covariates including troponin in predicting mortality" in the Discussion section, might be supported with more specific data.

Minor comments:

13. The emphasis on "mildly elevated hsCRP" in describing association with mortality risk might be clarified; from Figure 2, it appears that the association exists over a large range of hsCRP elevation.

14. Assessment of hsCRP independent of troponin level is also variously referred to as "beyond troponin" (in the title) and "above troponin" (in the abstract). It might be considered to standardize the terminology throughout, for clarity.

15. In the Statistical Analysis section, it is stated that "the IDI reflects the change in calculated risk for each individual"; it might be clarified as to whether this change is positive by default (i.e. improvement).

Reviewer #2: This paper describes the incremental long-term prognostic value of high-sensitivity C-reactive protein (hsCRP) above troponin in a large real-world cohort of unselected ACS patients. The objective was to determine if mildly elevated hsCRP was associated with mortality risk, irrespective of troponin level. Eligible patients from 2010 until 2017 were included retrospectively, utilizing NIH Research Health Informatics Collaborative data collected from five hospital centers in Great Britain. Patient selection was based on troponin measurements, normal white blood cell counts and a hsCRP up to 15 mg/L. Patients were followed up until death or censoring within 3 years. To standardize, the ratio of the observed peak troponin value was divided by the ULN for each troponin assay. 102337 patients were included in the analysis. The population was divided into four categories, those with a normal hsCRP <2mg/L, and three groups of incrementally increasing hsCRP (2-4.9 mg/L, 5-9.9 mg/L and 10-15 mg/L). On multivariable Cox regression analysis, there was a positive and graded relationship between hsCRP level and mortality at baseline, which remained at three-years, irrespective of troponin values. 

The background for this mega-study is based on the hypothesis that inflammation is a contributory factor for the development of atherosclerosis, supported by RCTs, such as CANTOS, COLCOT and LoDoCo2, the former applying canakinumab and the two latter applying colchicine as anti-inflammatory agents, whereas no effect of methotrexate was obtained in CIRT. These trials were performed in patients with established coronary heart disease, with cardiovascular disease as a primary endpoint, reflecting progression of atherosclerotic disease, whereas the current study presents results related to total mortality, which also includes non-cardiac death, and, ideally, this should have been accounted for when discussing the significance of hsCRP as a marker of atherosclerotic disease. 

As patients in the highest as compared to the lowest quartile of hsCRP are 10 years older, one would expect an increasing proportion of co-morbidity through the quartiles of this observational study. These limitations are essential, especially when relating the results to progressive atherosclerotic disease and recommending hsCRP for selection of high-risk patients that may benefit from anti-inflammatory treatment. 

Hs-CRP is an acute phase reactant and may not be a reliable indicator of risk, when not measured during steady state. However, the observation that mortality was greatest in patients with raised hs-CRP and moderate troponin elevations, might support the authors' claim.

In short, discussion and limitations require more attention.

Reviewer #3: This manuscript describes the importance of measuring hsCRP in ACS patients as prognostic factor in a large patient cohort. This research question is crucial to further provide information on patient selection for anti-inflammatory therapies and early prognosis of deleterious outcomes. 

The main claims are that CRP admission levels of suspected and actual ACS patients significantly correlate with 30-day and 3-year mortality of these patients. It is therefore concluded that CRP should be routinely assessed in ACS patients and not discarded as only a systemic inflammation marker. Together with troponin, risk assessment of ACS patients could be easily improved. 

Minor points:

1) The authors should discuss critically and in more detail that all data analysis is based on the initial hsCRP value at hospital admission. They write "that our hsCRP and troponin values represent presentation values rather than sequelae of the index event". The authors should put this in the context of recent publications regarding CRP's active role in the pathophysiology of acute coronary incidents such as AMI and the acute phase response in general. Is there information of the exact time point of CRP measurement and actual diagnosed ACS? This could enlighten whether it is actually the CRP baseline value or already the initial increase of the acute phase response. By excluding patients with CRP > 15 mg/L not only infections but also already ongoing acute coronary events were excluded. The authors also do not mention that in the CAMI-1 trial, a significant association was found between CRP gradient (CRP increase between 12 and 32 hours after symptom onset) and infarct size, LVEF, and circumferential and longitudinal strain. In addition, some publications reported a significant correlation between infarct size and peak CRP, which could also be used as a predictor (most from University of Innsbruck). Both could be discussed in relation to the use of CRP level on admission. In the context of a 2016 meta-analysis showing a significant association between infarct size and mortality (Stone et al.), another point could be discussed.

2) Given the strong evidence for the pathophysiological effect of a high CRP level, the following statement in the manuscript could be supplemented by citing recent literature: "The striking finding that patients in the highest hsCRP group, but who were troponin-negative, had similar mortality to those who were troponin-positive but had hsCRP<2 mg/L, brings into question whether, in clinical practice, we are missing high-risk patients who may benefit from further investigation and treatment."

3) Following this, the authors should also introduce CRP not only as end-protein read-out of the IL-6 cascade. They write this in the context of atherosclerosis and baseline CRP levels but should include CRP's pathophysiological role during and after acute incidents.

4)The statistical methods are well described and mostly appropiate. Among others, the continuous NRI and IDI are used to show the improvement in the prediction model, when hsCRP and troponin positivity are added and p-values are calculated. Continuous NRI and IDI are frequently used, but also prone to error, especially if nested models are used and p-values are calculated (Pepe et al., 2014, Pencina et al., 2015, Burch et al. 2017). The continuous NRI and IDI could be used to describe the improvement of the model, but likelihood-based methods should be used to calculate p-values (Pepe et al. 2013, Burch et al. 2017).

Since there are already significant differences in the AUROC for the different models and there is a clear improvement in NRI and IDI, it is very likely, that the likelihood-based calculation will support the improvement of the models.

Literature:

Pepe et al. 2013: Testing for improvement in prediction model performance, Stat Med. 2013 Apr 30; 32(9): 1467-1482.

Pepe et al. 2014: Net Risk Reclassification P Values: Valid or Misleading?, J Natl Cancer Inst. 2014 Apr; 106(4): dju041.

Pencina et al. 2015: RE: net risk reclassification P Values: valid or misleading?, J Natl Cancer Inst. 2014 Nov 27;107(1):355.

Burch et al. 2017: Net Reclassification Index and Integrated Discrimination Index Are Not Appropriate for Testing Whether a Biomarker Improves Predictive Performance, Toxicol Sci. 2017 Mar; 156(1): 11-13.

[LINK]

---

## [Decision Letter · Decision Letter 2]

23 Dec 2021

Dear Dr. Mayet,

Thank you very much for re-submitting your manuscript "The mortality risk prediction of mildly elevated high-sensitivity C-reactive protein beyond troponin in 102337 patients with suspected acute coronary syndrome (NIHR Health Informatics Collaborative CRP-RISK Study): A retrospective cohort study" (PMEDICINE-D-21-03948R2) for review by PLOS Medicine.

I have discussed the paper with my colleagues and the academic editor and it was also seen again by two reviewers. I am pleased to say that provided the remaining editorial and production issues are dealt with we are planning to accept the paper for publication in the journal.

[LINK]

We hope to receive your revised manuscript within 2 weeks. Please email us (plosmedicine@plos.org) if you have any questions or concerns.

We look forward to receiving the revised manuscript by Jan 06 2022 11:59PM.   

Sincerely,

Callam Davidson, 

Associate Editor 

PLOS Medicine

plosmedicine.org

Requests from Editors:

Please update your line numbering such that it is continuous throughout the document (i.e. does not restart with each new page).

Please update your full title to ‘Mortality risk prediction of high sensitivity C-reactive protein in suspected acute coronary syndrome: a cohort study’.

Please trim your abstract as it is currently slightly too long. Key information to include: study design, population and setting, number of participants, years during which the study took place, length of follow up, main outcome measures, quantified main results (with 95% CIs and p values), and important dependent variables adjusted for in the analyses. Additional methodological detail (e.g. detailed exclusion criteria [lines 14-16], standardisation of troponin levels [lines 17-18], exposure classification [lines 18-19]) can be removed. Additionally, you can remove the content from ‘All analyses on hsCRP…’ to ‘…using the peak troponin level’ in paragraph two of the abstract methods and findings (lines 24-27). 

Please also condense the ‘limitations’ section of the abstract (lines 10-13) such that it is only a single sentence beginning ‘The main limitations of the study are’, or similar. 

Given the observational nature of your study, I suggest updating the first sentence of the abstract conclusions to read ‘These multi-centre, real-world data from a large cohort of patients with suspected ACS suggest mildly elevated hsCRP (up to 15 mg/L) may be a clinically meaningful prognostic marker beyond troponin and point to its potential utility in selecting patients for novel treatments targeting inflammation.’

The Data Availability Statement (DAS) requires revision. If the data are not freely available, please describe briefly the ethical, legal, or contractual restriction that prevents you from sharing it. Please also include an appropriate contact (web or email address) for inquiries (this cannot be a study author). 

Please update your Author Summary using the standard PLOS Medicine structure. In brief, summaries should comprise of 2-3 single sentence bullet points for each of the three questions. See full guidance here (https://journals.plos.org/plosmedicine/s/revising-your-manuscript) and consult previous PLOS Medicine articles for examples (https://journals.plos.org/plosmedicine/). 

Page 8, line 12: Please cite specific items within the supplementary material rather than citing the entire section (e.g. S1 Analysis Plan). See our website for more detailed guidance (https://journals.plos.org/plosmedicine/s/supporting-information).

Please add the following statement, or similar, to the Methods: "This study is reported as per the Strengthening the Reporting of Observational Studies in Epidemiology (STROBE) guideline (S1 Checklist)."

Did your study have a prospective protocol? Please state this (either way) early in the Methods section.

Thank you for providing your prospective analysis plan. Changes in the analysis-- including those made in response to peer review comments-- should be identified as such in the Methods section of the paper, with rationale.

Page 18, line 5: Please remove the citation for Supplementary Figure 2.

References: Please ensure all journal names are consistently abbreviated using the abbreviations found in the National Center for Biotechnology Information (NCBI) databases.

Please define the statistical test used to derive the p values in the legend of Table 2.

You allude to statistical testing to determine baseline differences between participants; however no p values are reported in Table 1. Please update Table 1 to include relevant p values and report the statistical test used in the legend.

Thank you for providing your STROBE checklist. Please replace the page numbers with paragraph numbers per section (e.g. "Methods, paragraph 1"), since the page numbers of the final published paper may be different from the page numbers in the current manuscript.

The terms gender and sex are not interchangeable (as discussed in http://www.who.int/gender/whatisgender/en/ ); please use the appropriate term.

Comments from Reviewers:

Reviewer #1: We thank the authors for addressing our previous concerns. However, the previous supplementary figure on Kaplan-Meier curves by hsCRP level and ACS diagnosis appears to have been removed; was there any reason for this?

Reviewer #2: The authors have addressed my concerns. I have no further comments.

[LINK]

---

## [Editor Report · Decision Letter 3]

11 Jan 2022

Dear Dr Mayet, 

On behalf of my colleagues and the Academic Editor, Dr Sanjay Basu, I am pleased to inform you that we have agreed to publish your manuscript "Mortality risk prediction of high sensitivity C-reactive protein in suspected acute coronary syndrome: a cohort study" (PMEDICINE-D-21-03948R3) in PLOS Medicine.

When making these formatting changes, please also update the following:

* Line 100: Correct the spelling of 'accurately'.

* Line 106: Please update to 'In this study of 102,337 patients with suspected heart attack, a higher hsCRP level was associated with a higher risk of death'.

* Line 113: Please update to 'These findings suggest hsCRP is a clinically meaningful marker of risk of death in addition to troponin in patients with suspected heart attack'.

PRESS

Sincerely, 

Callam Davidson 

Associate Editor 

PLOS Medicine